# Detection of Oxytetracycline Using an Electrochemical Label-Free Aptamer-Based Biosensor

**DOI:** 10.3390/bios12070468

**Published:** 2022-06-28

**Authors:** Sanaz Akbarzadeh, Habibollah Khajehsharifi, Saeedeh Hajihosseini

**Affiliations:** 1Department of Chemistry, Faculty of Science, Yasouj University, Yasouj 75918-74831, Iran; akbarzade.sanaz@yahoo.com; 2Medical Nanotechnology and Tissue Engineering Research Science Institute, Shahid Sadoughi University of Medical Science, Yazd 8919-5999, Iran; s.hosseini.ra@gmail.com

**Keywords:** oxytetracycline, aptasensor, nanocomposite, differential pulse voltammetry, electrochemical impedance spectroscopy

## Abstract

One of the most effective ways to detect and measure antibiotics is to detect their biomarkers. The best biomarker for the control and detection of oxytetracycline (OTC) is the OTC-specific aptamer. In this study, a novel, rapid, and label-free aptamer-based electrochemical biosensor (electrochemical aptasensor) was designed for OTC determination based on a newly synthesized nanocomposite including multi-walled carbon nanotubes (MWCNTs), gold nanoparticles (AuNPs), reduced graphene oxide (rGO), and chitosan (CS), as well as nanosheets to modify a glassy carbon electrode, which extremely enhanced electrical conductivity and increased the electrode surface to bind well with the amine-terminated OTC-specific aptamer through self-assembly. The (MWCNTs-AuNPs/CS-AuNPs/rGO-AuNPs) nanocomposite modified electrode was synthesized using a layer- by-layer modification method which had the highest efficiency for better aptamer stabilization. Differential pulse voltammetry (DPV), cyclic voltammetry (CV), electrochemical impedance spectroscopy (EIS), and scanning electron microscopy (SEM) techniques were used to investigate and evaluate the electrochemical properties and importance of the synthesized nanocomposite in different steps. The designed aptasensor was very sensitive for measuring the OTC content of milk samples, and the results were compared with those of our previously published paper. Based on the calibration curve, the detection limit was 30.0 pM, and the linear range was 1.00–540 nM for OTC. The repeatability and reproducibility of the aptasensor were obtained for 10.0 nM of OTC with a relative standard deviation (RSD%) of 2.39% and 4.01%, respectively, which were not affected by the coexistence of similar derivatives. The measurement in real samples with the recovery range of 93.5% to 98.76% shows that this aptasensor with a low detection limit and wide linear range can be a good tool for detecting OTC.

## 1. Introduction

Aptamers are single-stranded oligonucleotide ligands (including DNA and RNA) that are between 30 and 70 nucleotides long. These single-strand sequences are twisted and bind specifically to target substances [1]. Binding to the target occurs through hydrogen bonds, electrostatic reactions, weak van der Waals forces, or combinations of them [2]. Aptamers are typically developed in vitro with a process called the systematic evolution of ligands by exponential enrichment (SELEX) and can be carefully designed to bind exclusively to various targets such as proteins, peptides, enzymes, cell surface receptors, microorganisms, etc. It can also be a proper substitute for an antigen or antibody. They can be attached to the surfaces of various inorganic substances for biosensor applications [3]. Some of the most important advantages of aptamers compared to an antigen/antibody include ease of modification and stabilization [4], robustness against heat [1], great selectivity for the target molecule [5], stability even under small pH changes and salt concentration [3], very small size [6], ease of synthesis and long-term storage [7]. The aforementioned triumphs of aptamers have led to their applications in various fields of therapeutics and diagnostics and eventually, their implementation into biosensor devices [6]. Termed aptasensors are designed to measure target molecules down to the nanomolar range [8]. So far, aptasensors have been mainly introduced in two categories: electrochemical and optical [8,9,10]. In the manufacture of electrochemical aptasensors, an electrode surface is usually used as a conductive surface to stabilize a biologically sensitive aptamer. The function of an electrochemical aptasensor is based on changes in the electrochemical current after the target molecule interacts with the aptamer [11].

OTC is an antibacterial drug that treats infections caused by bacteria. This medicine is also prescribed for some bacterial animal diseases [12]. The structure of OTC can be seen in Figure 1. Nowadays, excessive consumption of referees in livestock, which is performed for the treatment, prevention, and control of various diseases, poses irreparable dangers to the final consumer [13]. One of the most common drugs is antibiotics, which, if taken in excess, can leave antibiotic residues in meat, milk, and eggs. Foods contaminated with antibiotics, including tetracyclines, pose a serious threat to consumer health and, in some cases, even lead to increased general bacterial resistance in the human body [14]. In order to prevent the negative effects of antibiotics, most countries have set maximum residual levels (MRLs) for them. The European Union has identified MRLs for some antibiotics. For example, the MRL for penicillin in milk is reported to be 4.0 micrograms per kilogram [15]. There are several methods for determining the amount of antibiotic residue in food. Most of these methods are microbial inhibitory tests, also known as screening tests. These methods include the growth of bacteria such as Bacillus stearothermophilus; if there are antibiotics in milk, its growth is reduced or stopped [16]. Bacterial inhibitory tests are time-consuming and have no economic justification. Other methods of identifying antibiotics include liquid chromatography, gas chromatography, high-performance liquid chromatography, and mass spectrometry [17]. Although these methods are accurate, they require bulky and expensive equipment, complex sample preparation, an expert operator, and long-term incubation [18].

One method that does not have the abovementioned limitations uses a specific aptasensor to detect antibiotics [19]. In recent years, there has been an increased interest in using nanomaterials to enhance the sensitivity of aptasensors. Nanomaterials including gold nanoparticles and reduced graphene oxide are appropriate due to their acceptable electrical conductivity, and mechanical and thermal activities to be used in electrochemical biosensors such as aptasensors to modify the surface of the electrode [20]. On the other hand, gold nanoparticles are ideal for detecting different analytes and improving the conductivity of the aptasensor due to their unique structure, good electrical conductivity, and catalytic properties [21]. In order to improve the aptamer chains on the surface, chitosan is a proper material as it shows excellent filmmaking ability, relatively good conductivity, and sensitivity to chemical modifications, biocompatibility, and non-toxicity properties. Compared to nanomaterials, chitosan has weak electrochemical properties. Therefore, it is usually modified with nanomaterials such as AuNPs to improve its electrochemical properties and surface area [22]. So far, several studies have reported the determination of OTC antibiotic residues in food, such as Zhou et al. (2012), Meng et al. (2017), and Liu et al. (2017), [5,23,24]. In the aptasensor measuring system, the working electrode plays the main role because all the desired reactions occur on its surface. Different working electrodes have been introduced for application in electrochemical biosensors, including glassy carbon electrodes, different types of screen-printed electrodes, gold electrodes, platinum and carbon electrodes, as well as graphite. pencil core electrodes [25,26,27,28].

In this work, a glassy carbon working electrode was modified with reduced graphene oxide (rGO), multi-walled carbon nanotubes (MWCNTs), chitosan (CS), and gold nanoparticles (AuNPs) to increase its surface area and electrical conductivity. Additionally, a specific OTC aptamer was used to bind with OTC at the electrode surface. The interaction between the OTC and the stabilized aptamer on the surface of the aptasensor changed the oxidation current of the iron detector, (Fe^3+^/Fe^2+^), which was used as an electrochemical signal to measure OTC concentration.

## 2. Materials and Methods

### 2.1. Chemicals

Reagents and chemicals were all of analytical purity and provided from Merck (Darmstadt, Germany) and Sigma-Aldrich (Burlington, MA, USA) companies and used with no more purification. Also, double distilled water was used to prepare and dilute the solutions. MWCNTs was purchased from Sigma-Aldrich Company (Burlington, MA, USA) with the following characteristics: purity of more than 90%, an outer diameter of 70–110 nm, and an average length of 5–9 μm. An OTC amino DNA aptamer with the sequence of (5′-Amino-(CH_2_)_6_- CGT ACG GAA TTC GCT AGC CCC CCG GCA GGC CAC GGC TTG GGT TGG TCC CAC TGC GCG TGG ATC CGA GCT CCA CGT G-3) was purchased from TAG (Copenhagen, Denmark). The stock aptamer solution was created in a 0.1 M phosphate buffer solution at (pH = 7.0) containing 0.1 M of KCl. The iron detector solution contained 3.0 mM of K_4_Fe (CN)_6_/K_3_Fe (CN)_6_ and 0.1 M of KCl from Merck company (Darmstadt, Germany).

### 2.2. Apparatus and Electrodes

Voltammetric studies were performed using the AUT41203 potentiostat-galvanostat apparatus equipped with NOVA 2.1 software created by Metrohm/Autolab (EcoChemic, Utrecht, The Netherlands). A Metrohm pH meter 827, created by Metrohm/Autolab Company (Utrecht, The Netherlands), a Sartorius scale model BP221S, created by Sartorius Company (Göttingen, Germany) and a Pars Nahad ultrasonic bath model PARSONIC2600, created by Pras Nahad Company (Isfahan, Iran) and a centrifuge model EBA20 (Hettich, Germany) were used. Scanning electron microscope (SEM), XL30 Philips SEM (Hillsboro, OR, USA) was used to get the images of the electrode surfaces. All voltammetric measurements were performed with a three-electrode system including a glassy carbon electrode (GCE) as a working electrode, platinum (Pt) as an auxiliary electrode and silver/silver chloride (Ag/AgCl) as a reference electrode.

### 2.3. Synthesis of Nanocomposites

According to the Hummer method, graphene oxide (GO) was obtained from graphite powder. Additionally, reduced graphene oxide and gold nanoparticle nanocomposites (rGO-AuNPs) were synthesized based on the method proposed by Wang et al. [29] with minor modifications. For this purpose, 50.0 mg of graphene oxide was added to 20 mL of double-distilled water and placed under ultrasound waves for 2 h, and then 800.0 mg of polyvinylpyrrolidone (PVP) was added to the mixture and stirred for 12 h. In the next step, 250 µL of (1% *w*/*v*) HAuCl_4_ and 200.0 mg of ascorbic acid were added to the solution. Then, this solution was stirred for an hour at 95 °C. The resulting solution was centrifuged for 10 min, and, in order to remove excess PVP and ascorbic acid, the precipitated product was washed several times with water. Finally, the synthesized substances (rGO-AuNPs) were dispersed in 20 mL of water.

A chitosan and gold nanoparticle nanocomposite (CS-AuNPs) was synthesized according to the method presented by Sun et al. [30]. A mixture of 20 mL of CS in 2.0 M acetic acid (1% *w*/*v*) and 250 µL of HAuCl_4_ (1% *w*/*v*) was stirred for an hour, and then 100 μL of sodium borohydride (NaBH4) 0.4 mM was gradually added. The solution was stirred for an hour, and finally, the resulting ruby-red color solution indicated the successful synthesis of AuNPs.

For the synthesis of the MWCNT-AuNPs nanocomposite, in the first step, the primary samples were refluxed for approximately 15 h in 0.2 M of nitric acid solution. To remove metal impurities, create hydrophilicity, and functionalize the MWCNTs under these conditions, the surface of the MWCNTs was oxidized to carboxylic or ketone groups. The obtained materials were then thoroughly washed to remove residual acid and dried at room temperature. Next, the method proposed by Suresh et al. [31] was carried out with minor changes. A total of 0.05 g of functionalized MWCNTs was added to 5 mL of double-distilled water and the solution was exposed to ultrasound for an hour. Then 250 µL of HAuCl_4_ (*w*/*v* 1%) was added to the solution and stirred for an hour. At the end, 100 µL of (NaBH_4_) 0.4 mM was added to the solution, and, after stirring for an hour, MWCNTs-AuNPs were synthesized.

### 2.4. Fabrication of Nanocomposite Modified GCEs

According to our previous reports, the glassy carbon electrode was polished using 0.05 μm alumina slurry to result in a glossy surface and then washed with double distilled water three to five times [32]. In order to stabilize the synthesized (MWCNTs-AuNPs), (CS-AuNPs), and (rGO-AuNPs) nanocomposites on the glassy carbon electrode, 5 μL of each solution was placed on the surface of the polished electrode, and, after drying, it was washed with double-distilled water several times to remove unabsorbed nanocomposites from the surface. Finally, the electrode surface was allowed to dry [33].

In order to compare the results of different modified electrodes and the synergistic effect of nanocomposite components toward the determination of OTC and the peak current of the iron detector, two different modified electrodes including MWCNT-AuNP and MWCNT-AuNP/CS-AuNP composites were fabricated using the same procedure above.

### 2.5. Preparation of Aptasensor

The modified electrodes with the synthesized nanocomposites (MWCNTs-AuNPs/CS-AuNPs/rGO-AuNPs) were placed in a 0.1 M phosphate buffer solution (pH = 7.0) containing the aptamer with the optimum concentration and time of 20.0 nM for 10 h and then washed with double-distilled water and phosphate buffer solution to remove excess aptamer molecules and left to dry [33]. The aptasensors were each immersed separately for 30 min in a solution of 0.06 M phosphate buffer (pH = 7.0) and double-distilled water in the presence and absence of OTC at specific concentrations. It was washed and transferred to an electrochemical cell containing 10 mL of iron detector solution, and its differential pulse voltammograms were taken in the range of −0.50 to 0.55 V. The resultant current of the iron detector in the solution containing OTC at the surface of the aptasensor is considered as an analytical signal. Finally, the Ip curve was plotted toward the OTC concentration [34]. The construction and detection processes are summarized in Figure 2.

### 2.6. Real Sample Preparation

The spiking method was applied to two milk samples to measure the OTC concentration in the milk samples. Then, the recovery was measured to confirm the satisfactory performance of the prepared electrochemical aptasensor. A total of 2 mL of milk samples was spiked with 10.0 nM, 20.0 nM, 25.0 nM, 50.0 nM, and 100 nM of OTC and then diluted by 4 mL of ultrapure water. Subsequently, 2 mL of chloroform and 2 mL of a 10% solution of trichloroacetic acid were added to the diluted samples, which were stirred for 2 min. Then, for 20 min, ultrasonic treatment was performed at 5000 rpm, the final solution was centrifuged for 10 min, and the resultant floated layer was used to determine OTC [35].

## 3. Results

### 3.1. Characterization of Nanocomposites

The morphology of the synthesized nanocomposites was determined by the SEM imaging method. Figure 1A–C show the SEM images of modified glassy carbon electrodes with MWCNTs-AuNPs, CS-AuNPs, and rGO-AuNPs, respectively. As shown in (Figure 1A), the MWCNTs were homogeneously distributed on the surface of the glassy carbon electrode, and the gold nanoparticles were properly visible in this image. (Figure 1B) shows the AuNPs with different diameters on the CS film. In (Figure 1C), the bright dots represent gold nanoparticles distributed all over the surface of the glassy carbon electrode, and most of their density is concentrated at the hedge sites and rGO edges. (Figure 1D) also shows the rGO-AuNP nanocomposite on the MWCNT-AuNP/Cs-AuNP nanocomposite on the surface of the glassy carbon electrode. The results of these images well indicate the proper placement and distribution of these synthesized nanocomposites on the surface of the glassy carbon electrode.

### 3.2. Cyclic Voltammetric and Impedance Measurements

To investigate the electrode surface modification process, cyclic voltammetry and impedance spectroscopy techniques were used. Cyclic voltammograms for the unmodified glassy carbon electrode, the modified electrodes with MWCNTs-AuNPs, MWCNTs-AuNPs/CS-AuNPs, and MWCNTs-AuNPs/CS-AuNPs/rGO-AuNPs were recorded in the iron detector solution. As seen in (Figure 2A), the unmodified glassy carbon electrode showed two clear oxidation-reduction peaks, and the modification of the glassy carbon electrode with the MWCNT-AuNP nanocomposite increased the oxidation-reduction peak current on the surface of the electrode compared to the unmodified electrode. In addition, the separation of the cathodic and anodic peaks was reduced owing to the enhancement in the surface area, and the electrical conductivity was increased due to the use of gold nanoparticles and MWCNTs.

To confirm the results of the cyclic voltammetry, electrochemical impedance spectroscopy was also carried out (Figure 2B). As shown in this figure, the comparison of the decrease in the charge transfer resistance of the MWCNT-AuNP nanocomposite-modified electrode with that of the unmodified electrode confirms the results of the cyclic voltammetry. Moreover, after adding 0.5 μL of CS-AuNPs to the MWCNT-AuNP-modified glassy carbon electrode, the current of the cathodic and anodic peaks of the Fe^3+^/Fe^2 +^ detector also increased (Figure 2A, curve (c)) due to more addition of gold nanoparticles at the electrode surface. (Figure 2B curve (c)) also demonstrates a decrease in the charge transfer resistance, indicating an increase in electrical conductivity. Finally, after adding rGO-AuNPs to MWCNTs-AuNPs/CS-AuNPs/GCE, a further increase in the cyclic voltammogram current (Figure 2A curve (d)) and a decrease in the charge transfer resistance of the electrochemical impedance (Figure 2B curve (d)) were observed. This intense increase in current was due to the cooperative effect of the good electrical conductivity of rGO and AuNPs.

### 3.3. Evaluation of the Aptasensor Performance

In order to investigate the performance of the MWCNTs-AuNP/CS-AuNP/rGO-AuNP nanocomposites in OTC measurement, the modified glassy carbon electrodes with different nanocomposites of MWCNTs-AuNPs, MWCNTs-AuNPs/CS-AuNPs, and MWCNTs-AuNPs/CS-AuNPs/rGO-AuNPs were prepared and compared with each other in the presence and absence of OTC. (Figure 3) shows the differential pulse voltammograms of these three modified glassy carbon electrodes with different surfaces for the detection of OTC. (Figure 3 ((a), voltammograms)) shows the differential pulse voltammograms of the unmodified glassy carbon electrodes in iron detector solution. As it can be observed, by modifying the glassy carbon electrode with these three modifiers, the oxidation peak current of the iron detector was increased (Figure 3 ((b), voltammograms)), which indicates the high electrical conductivity of the synthesized nanocomposites. While the aptamer was placed on the nanocomposite-modified electrodes in all states, the rate of decrease was different for each electrode. While the aptamer was placed on these nanocomposites, the oxidation peak current of the iron detector decreased (Figure 3 ((c), voltammograms)), which shows a decrease in electron transfer and the prevention of electrons from reaching the surface, while the rate of reduction in current depends on different electrode surfaces. As demonstrated in (Figure 3 ((d), voltammograms)), after the interaction of the aptamer with OTC for 30 min, the relating signal decreased again, which revealed the hindrance of the electron transfer of [Fe (CN)_6_]^3−/4−^ on the electrode surface. The results of the differential pulse voltammetry of these electrodes indicate that using an MWCNT-AuNP/CS-AuNP/rGO-AuNP nanocomposite due to the interactive effect of all components in active surface enhancement, electrical conductivity, and reduction in charge transfer resistance can cause a greater increase in the peak current of the iron detector at the surface of the electrode toward MWCNT-AuNP and MWCNT-AuNP/CS-AuNP nanocomposites. The value of (Ip) decreased after the stabilization of the OTC on the electrode. Therefore, the differential pulse voltammetry results show that MWCNTs-AuNPs/CS-AuNPs/rGO-AuNPs can be used as an excellent modifier for measuring OTC.

### 3.4. Optimization of Time and Concentration

Aptamer modification time and concentration are important in the modification procedure and are effective factors in the performance of aptasensors. The aptamer was stabilized on the electrode surface based on the self-assembly method. The modification procedure cannot be completed in a short time and may take a long time. The results of different aptamer modification times were compared and the greatest peak current was achieved at the modification time of 10 h (Figure 4A). Therefore, 10 h was chosen as the optimum aptamer modification time. The degree of stabilization of the aptamer on the electrode surface depends on the concentration of the aptamer used for modification and thus affects the amount of OTC which can bind to the aptamer and the resultant electrochemical signal of the prepared electrochemical aptasensor. At the concentration of 20.0 nM, the value of the peak current reached a steady stage, and increasing the aptamer concentration could barely change the peak current value (Figure 4B).

To investigate the effect of incubation time on the interaction between OTC and the aptamer, an aptasensor created with MWCNT-AuNP/CS-AuNP/rGO-AuNP GCE was placed in a 0.1 M phosphate buffer solution (pH = 7.0) containing 10.0 nM of OTC at different times of exposure. After washing with phosphate buffer and double distilled water to remove any unwanted absorbed reagents, the aptasensor was transferred to an electrochemical cell containing 10 mL of iron detector solution, and subsequently, the differential pulse voltammogram of the solution was recorded. The currents obtained from the redox reaction of the iron detector were used as the analytical signal of the aptasensors that interacted with OTC. Finally, the current curve was plotted versus time by measuring the currents at different times (Figure 4C). The results showed that by increasing the interaction time of OTC and the stabilized aptamer on the electrode surface, the iron oxidation peak current (Ip) decreased. Then, after 30 min, it reached the lowest level and stayed constant. Therefore, to draw the calibration curve, 30 min was considered the optimal time for OTC interaction with the aptamer.

The voltammograms obtained by the interaction of different concentrations of OTC with the stabilized aptamer are shown in (Figure 5A). By measuring the iron detector currents related to different OTC values, the relationship between the OTC concentration and the current was plotted to the corresponding curve (Figure 5B). This figure reveals a linear relationship between Ip versus the OTC concentration. By increasing the concentration of OTC, there was an increase in the amount of recognized OTC at the surface of the aptasensor (Figure 5A), which prevented the electron transfer of [Fe (CN)_6_]^3−/4−^ on the surface of the modified GCE and caused a decrease in the DPV Ip current value. According to (Figure 5B), the relation between the analytical signal and the concentration was linear in the range of 1.00–540 nM.

The detection limit of this aptasensor was calculated by the equation LOD = 3*Sb*/*m*, in which *Sb* is the standard deviation of the blank signal, and *m* is the slope of the calibration curve. Therefore, under optimal circumstances, five different aptasensors were placed in the control solution for 30 min without OTC. Then, the corresponding standard deviation was obtained by measuring the current of each aptasensor separately in the iron detector solution. The detection limit of the method for measuring OTC was 30.0 pM.

### 3.5. Investigation of Aptasensor Repeatability and Reproducibility

To investigate the repeatability of the fabricated sensor’s response to the OTC measurement, the Ip signals of the 10.0 nM OTC measurement were recorded by five separate aptasensors (which had been constructed similarly). The percentage of the standard deviation of the obtained currents was 2.39, which is proof of good repeatability of the response of these aptasensors. The aptasensor reproducibility response was evaluated by recording the obtained signals from 10.0 nM of OTC from five separate aptasensors on five different days. The relative standard deviation of the measured currents was 4.01%, which indicates the good reproducibility of this aptasensor.

### 3.6. Evaluation of Aptasensor Sustainability

The evaluation of the sustainability of the prepared aptasensor was performed by five fabricated aptasensors, which were kept at 4 °C for 30 days and then used to measure a concentration of 10.0 nM of OTC. The results showed that the signal decreased by 5.96% after 30 days of storage of the aptasensors under optimum conditions, indicating good stability and long life of the aptasensor.

### 3.7. Interference Study

The results of the interference study on the analytical signal of OTC concentration of 10.0 nM and the structural analogues’ concentration of 50.0 nM are shown in (Figure 6), which, from left to right, depicts OTC without interference (OTC), OTC in the presence of doxycycline (DOX), OTC in the presence of chlortetracycline (CTC), and OTC in the presence of tetracycline (TET). The coexistence of the interfering substances and OTC at the same concentration could not significantly change the electrochemical aptasensor Ip current values. The results in (Figure 6) show good selectivity of the proposed aptasensor for OTC detection. Therefore, adding these species at a concentration of 50.0 nM does not interfere with the OTC measurements.

### 3.8. Real Sample Evaluation

The evaluation of the aptasensor performance for the determination of OTC was completed in real samples. In order to determine the concentration of OTC in milk samples, the spiking method was used and the recovery was measured. The summarized results in Table 1 reveal that the average of the recovery range was 93.5–98.76%, with the RSDs lower than 3.44% using the proposed aptasensor. This shows that the determination of OTC via this aptasensor with acceptable recovery is possible in natural samples.

Our new proposed electrochemical aptasensor is comparable with previously reported works in analytical performance, and Table 2 shows the summarized comparison of the aspects of modification, linear range, and limit of detection.

## 4. Discussion

In this study, the amount of reduction in the oxidation peak current of Fe^3+^/Fe^2+^ detector after the interaction of OTC with the aptamer attached to the electrode surface was considered as a signal. A great signal reduction was observed when all the MWCNT-AuNP, rGO-AuNP, and CS-AuNP modifiers were stabilized on the electrode surface. In other words, the modified electrode with the MWCNT-AuNP/CS-AuNP/rGO-AuNP nanocomposite proved a suitable substrate for further aptamer stabilization and consequently more OTC interaction with aptamer, which was due to the cooperative effect of rGO, MWCNTs, Cs, and AuNPs in the enhancement of the electrode surface and increase in electric conductivity, as well as electrocatalytic properties [41]. This cooperative effect increases sensitivity and efficiency and improves the electrochemical signal in OTC measurements. The high sensitivity of this method is due to the properties of nanocomposite constituents. MWCNTs increase the surface-to-volume ratio, electrical conductivity at the electrode surface, electron transfer kinetics and specific regions for efficient stabilization of large amounts of DNA aptamer while maintaining its biological activity [30]. AuNPs are also used due to their excellent surface-to-volume ratio, which helps improve conductivity between rGO sheets, while rGO further stabilizes AuNPs [42]. CS also provides a suitable substrate for better and more aptamer stabilization due to its high adhesion properties, functional groups which strongly interact with the amine-terminated groups of aptamers, biocompatibility, good relative conductivity, and excellent filmmaking ability. The presence of CS in the MWCNT-AuNP/CS-AuNP/rGO-AuNP nanocomposite increased the reproducibility of this aptasensor. Erdem et al. [43] showed that high adhesion and the excellent filmmaking capabilities of CS caused repeatable films on the electrode surface. The combination of these properties in the constituents of the nanocomposite results in excellent electrocatalytic activity and effective signal amplification [44]. The persistence of the analytical signal after 30 min of interaction between OTC and stabilized aptamer on the electrode surface indicated that the active aptamer sites had reached saturation and the maximum interaction had taken place [41]. The detection limit is one of the most important figures of merit in comparing different methods of analysis. Although our group’s previously designed electrochemical sensor for OTC detection had a wider linear range than the aptasensor presented in this study, this new aptasensor has a lower detection limit of 30.0 pM.

In this study, the presence of mixed CS with AuNPs in the middle layer, between rGO-AuNPs and MWCNTs-AuNPs, caused a decrease in the percentage of relative standard deviation (RSD%) to 2.39% for five repetitive measurements. One of the limitations of this study was that this aptasensor was non-portable. Therefore, further studies are needed to stabilize the MWCNT-AuNP/CS-AuNP/rGO-AuNP nanocomposite and OTC aptamer on portable electrical kits under completely optimum and safe conditions to make a suitable biochip.

## 5. Conclusions

In this research, a rapid and extremely sensitive electrochemical label-free aptamer-based biosensor was fabricated to measure and detect OTC. It was designed with an OTC-specific aptamer as a biomarker and an MWCNT-AuNP/CS-AuNP/rGO-AuNP nanocomposite as a modifier to modify the glassy carbon electrode surface. Cyclic voltammetry, differential pulse voltammetry, and impedance spectroscopy techniques were performed to investigate the electrochemical properties of the proposed aptasensor. The results of our study show that the nanocomposite-modified electrode (MWCNTs-AuNPs/CS-AuNPs/rGO-AuNPs GCE) had the highest efficiency in better aptamer stabilization. Also, the importance of the nanocomposite in OTC measurement was evaluated by the differential pulse voltammetry technique, which showed a cooperative effect between MWCNT-AuNP, CS-AuNP and rGO-AuNP nanocomposites.

## Data Availability

Not applicable.

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
