# Peer review of "Detection of Oxytetracycline Using an Electrochemical Label-Free Aptamer-Based Biosensor"

_biosensors, 2022, doi:10.3390/bios12070468_

Round 1

Reviewer 1 Report

This paper is focused on designing an electrochemical aptasensor to measure Oxytetracycline. An electrode prepared with multi-walled carbon nanotubes, gold nanoparticles, reduced graphene oxide, and chitosan nanosheets was used for this porpoise. Although the approach is interesting, there is no novelty in the results. Supplementary Table S1 should be mentioned in the main text and analytical parameters compared with similar developments already published. When analyzing milk samples, more concentrations should be studied if the linear range is 1.00 - 540 nM, only concentrations of 20,25 and 50 nM were tested. English should be improved. For all the above, this work is not suitable for being published.

Author Response

Dear Reviewer

We are grateful to you for your insightful comments on our manuscript. Absolutely your comments are so valuable and constructive and can help us to level up our paper. We have been able to incorporate changes to reflect most of the suggestions provided by you. All the changes are highlighted. Here is a point-by-point response to your comments and concerns.

We all tried to apply all the revisions and comments in English improvement which is revised and highlighted in yellow.

-The relevance of references to the text is checked and highlighted in green.

-As you asked for mentioning of Table S1 in the main text the name of the Table is changed to Table 2 in page 20.

-Two more concentrations of OTC in milk samples were studied and the results were added to the Table1 to show the capability of the presented aptasensor for the determination of OTC in the whole linear range on page 19.

Thanks for pointing out that the approach of our work is interesting. In this work we tried to introduce a new aptasensor and enhanced the ease of use, sensitivity and selectivity of the proposed aptasensor using specific OTC aptamer bounded to a new nanocomposites of MWCNTs-AuNPs/CS-AuNPs/rGO-AuNPs which could enhance the detection limit of the sensor, to 30 pM which is comparable to the results of our previous published paper and some other papers which are listed in Table2. The distinctive features of our proposed aptasensor are simple operation, low cost and wide detection linear range, high throughput analysis and fast response.

Reviewer 2 Report

This manuscript has been designed an aptamer-based electrochemical label-free biosensor (electrochemical aptasensor) to measure OTC. However, some revisions should be made to make the paper more convincing.

1) What is the innovation of this manuscript? Please include it in the abstract.

2) The abbreviation "OTC" and the full name "oxytetracycline" should be unified throughout the manuscript.

3) The binding principle of aptamer at the sensing interface should be carried out.

4)  Fe+3/Fe+2 should be revised with Fe3+/Fe2+.

5) The references have some formatting problems, such as the page numbering of Reference 3, 4, 5, 6, 7, 8, 9, 10, 11, 31, 32, 34, 37, 38, 39; “-” should be revised “–”.

6) The authors should pay attention on the quality of figures, and the page number is needed.

7)  There are some typos in the manuscript, please carefully check the text.

Author Response

Dear Reviewer

We are grateful to you for your insightful comments on our manuscript. Absolutely your comments are so valuable and constructive and can help us to level up our paper. We have been able to incorporate changes to reflect most of the suggestions provided by you. All the changes are highlighted. Here is a point-by-point response to your comments and concerns.

We all tried to apply all the revisions and comments in English improvement which is revised and highlighted in yellow.

1) What is the innovation of this manuscript? Please include it in the abstract.

In this study we tried to present a highly sensitive and rapid label free aptasensor with new approach of modification through different nanocomposites to reach to a very low detection limit of 30 pM, by using MWCNTs-AuNPs, MWCNTs-AuNPs/Cs-AuNPs and MWCNTs-AuNPs/Cs-AuNPs/rGO-AuNPs. Comparison between the results of different modified electrodes toward the determination of OTC showed that cooperative effect of MWCNTs-AuNPs/Cs-AuNPs/rGO-AuNPs make a proper bound with amine terminated OTC aptamer through self-assembly method and showed the highest conductivity and sensitivity. The distinctive features of our proposed aptasensor are simple operation, low cost and wide detection linear range, high throughput analysis and fast response.

2) The abbreviation "OTC" and the full name "oxytetracycline" should be unified throughout the manuscript. 

Oxytetracycline is mentioned for the first time and the abbreviation “OTC” is substituted in throughout the whole manuscript and highlighted.

3) The binding principle of aptamer at the sensing interface should be carried out.

The aptamer was bounded to the nanocomposites through self- assembly method with the long modification time of 10 h which is mentioned in the abstract and also result and discussion sections. 

4)  Fe+3/Fe+2 should be revised with Fe3+/Fe2+.

It is corrected and highlighted in the text on page 6.

5) The references have some formatting problems, such as the page numbering of Reference 3, 4, 5, 6, 7, 8, 9, 10, 11, 31, 32, 34, 37, 38, 39; “-” should be revised “–”.

All the references are checked and formatted, some of them are changed and highlighted in green color.

6) The authors should pay attention on the quality of figures, and the page number is needed.

The page number is added and the quality of all the figures is improved.

7)  There are some typos in the manuscript, please carefully check the text.

All typos are corrected and highlighted in the text in yellow color.

Reviewer 3 Report

In the paper ‘Detection of Oxytetracycline Using an Electrochemical Label free Aptamer-Based Biosensor’ by Akbarzadeh S. et al. an aptamer base electrochemical sensors was evaluated for the detection of Oxytetracycline (OTC).

Before publication, I would recommend the authors to address the following points:

·        Figure 1 does not show any scale bar so it is hard to get an idea of the NPs dimension. I have also missed this information in the text.

·        The authors claim better performance of the electrode containing all the components, i.e. MWCNTs-AuNPs, rGO-AuNPs and CS-AuNPs, did you compared the glassy carbon electrode modified with each of the single components and the aptamer?

·        Could you comment about the re-usability of this sensor?

·        Did you perform any sample preparation prior the analysis of milk with the aptasensor?

·        What do you mean with ‘Recovery’ in Table 1?

·        There are several typos, example ‘’pea’’ instead of ‘’peaks’’ (page 7), on page 4 ‘’in’’ is repeated two times. On page two I do not understand the sentence ‘’ In order to prevent the negative effects of antibiotics, most countries have set maximum residual levels of MRLs.’’, you mean the maximum residual levels of antibiotics?

Kindly revise the manuscript one more time.

·        Many literature is missing: https://doi.org/10.1021/ac3037574, https://doi.org/10.1016/j.aca.2008.12.025, 

·        Related to the second citation I am referring here, what is your manuscript adding to the current state of the art?

Author Response

Dear Reviewer

We are grateful to you for your insightful comments on our manuscript. Absolutely your comments are so valuable and constructive and can help us to level up our paper. We have been able to incorporate changes to reflect most of the suggestions provided by you. All the changes are highlighted. Here is a point-by-point response to your comments and concerns.

We all tried to apply all the revisions and comments in English improvement which is revised and highlighted in yellow.

Figure 1 does not show any scale bar so it is hard to get an idea of the NPs dimension. I have also missed this information in the text.

The scale bar is added to all the images in Fig 1, page 12.

  • The authors claim better performance of the electrode containing all the components, i.e. MWCNTs-AuNPs, rGO-AuNPs and CS-AuNPs, did you compared the glassy carbon electrode modified with each of the single components and the aptamer?

The comparison of different modifications on the surface of electrodes were investigated and the results are added to the manuscript in Fig 3, page 15.

.        Could you comment about the re-usability of this sensor?

Thanks for pointing out this issu but in this work we didn’t focus on investigation of the reusability or regeneration ability of the presented aptasensor, although there are some references which claimed that their aptasensors are regenerative or reusable via different strategies such as temperature shock, storage in NaCl solution, acetic acid treatment and etc. We used chitosan as a component of the nanocomposite which doesn’t resist in acetic acid solution and it can cause a phase change to chitosan. On the other hand the target molecules are usually tightly attached on the electrode surface due to their strong affinity with electrode-confined aptamers, this binding process is also take place in the presence of saline solution such as NaCl or KCl, therefore application of NaCl doesn’t work for us in regeneration and cleavage of the binding target molecule to the aptamer and it makes the sensors difficult to regenerate. Absolutely this issue is interesting to us to be investigated more in our next survey with new strategy.

  1. H. Yang, D. M. Shi, S. M. Zhu, B. J. Wang, X. J. Zhang and G. F. Wang, ACS Sens., 2018, 3, 1368–1375
  2. S. Liu, Z. H. Zhang, M. Chen, H. Zhao, F. H. Duan, D. M. Chen, M. H. Wang, S. Zhang and M. Du, Chem. Commun., 2017, 53, 3941–3944.
  • Did you perform any sample preparation prior the analysis of milk with the aptasensor?

In page 11, material and method section, the subsection of 2.1 sample preparation method was demonstrated and highlighted.

  • What do you mean with ‘Recovery’ in Table 1?

The ratio of measured amount of OTC by the apatasensor to real spiked amount of OTC, which was added to the sample multiplied to %100. Recoveries range is between  93.5% to 98.76% with the RSDs lower than 3.44% using the proposed aptasensor, representing that the determination of OTC via this aptasensor with acceptable recovery is possible in real samples. In order to demonstrate feasibility of the presented aptasensor for practical application in real samples the recovery assay performed.

  • There are several typos, example ‘’pea’’ instead of ‘’peaks’’ (page 7), on page 4 ‘’in’’ is repeated two times.

All the typos are corrected and highlighted in the text.

On page two I do not understand the sentence ‘’ In order to prevent the negative effects of antibiotics, most countries have set maximum residual levels of MRLs.’’, you mean the maximum residual levels of antibiotics?

This sentence is corrected and highlighted in introduction section.

  • Many literature is missing: https://doi.org/10.1021/ac3037574, https://doi.org/10.1016/j.aca.2008.12.025, 
  • Related to the second citation I am referring here, what is your manuscript adding to the current state of the art?

Our work is based on using an amine-terminated aptamer which was bound to the nanocomposite modifier layer via the self-assembly method at the electrode surface, the stability of the amine group binding to the nanocomposite is more than thiol termination to Au. The thiol termination group of the aptamer is usually bound to the conventional gold surface based on the affinity of thiol group to the Au electrode. On the other hand MWCNTs-AuNPs/CS-AuNPs/rGO-AuNPs nanocomposite is a suitable substrate for further aptamer stabilization and consequently more OTC interaction with aptamer, which is due to the cooperative effect of rGO, MWCNTs and AuNPs in the enhancement of electrode surface and increase of electric conductivity and also electrocatalytic properties. Due to using this recognition layer our sensor is much more sensitive toward the determination of OTC and has a much lower detection limit of 30 pM and a wider linear range of 1.00 t0 540 nM. AuNPs provide high conductivity and also much more surface area for aptamer binding in comparison with gold electrode. The nanocomposite mediator improved the electron relay during the entire electron transfer process and the aptasensor response speed. The distinctive features of our proposed aptasensor are simple operation, low cost and wide detection linear range, and high throughput analysis.

Round 2

Reviewer 1 Report

The revised manuscript now meets the requirements of academic publishing in this journal so it is suitable for publication. 

Author Response

Dear Reviewer

Thanks for very much for your comment toward our manuscript. The language of the manuscript is revised by an English language specialist. Please check it out and if there is still any problem with the language let us know.

Thanks for Your Time

Reviewer 2 Report

The manuscript can be accepted for publication.

Author Response

Dear Reviewer

Thanks for your consideration and agreement to our manuscript. The language is checked by an English language specialist. Please checked it out and if there is any problem let us know.

Thank You in Advance

Reviewer 3 Report

The paper is suitable for publication but the novelty of your approach (I am referring to the last question I posed in the last round of review) should be highlighted even more.

Author Response

Dear Reviewer

Thank you so much for your agreement and confirming of our work for publication. We tried to show the distinct points of our work toward the mentioned reference more in the abstract and also the language of the manuscript is double checked with an English language specialist. Please check it out and if there is any problem with it let us know.

Thanks for Your Time